# Influence of HRGO Nanoplatelets on Behaviour and Processing of PMMA Bone Cement for Surgery

**DOI:** 10.3390/polym13122027

**Published:** 2021-06-21

**Authors:** Jaime Orellana, Ynés Yohana Pastor, Fernando Calle, José Ygnacio Pastor

**Affiliations:** 1Departamento de Ciencia de los Materiales and CIME, Universidad Politécnica de Madrid, E28040 Madrid, Spain; jaime.orellana@upm.es (J.O.); ypastor@ucm.es (Y.Y.P.); 2Facultad de Medicina, Universidad Complutense de Madrid, E28040 Madrid, Spain; 3Departamento de Ingeniería Electrónica and ISOM, Universidad Politécnica de Madrid, E28040 Madrid, Spain; fernando.calle@upm.es

**Keywords:** polymethylmethacrylate, highly reduced graphene oxide, mechanical behaviour, bone cement, microstructure, surgery

## Abstract

Bone cement, frequently based on poly (methyl methacrylate), is commonly used in different arthroplasty surgical procedures and its use is essential for prosthesis fixation. However, its manufacturing process reaches high temperatures (up to 120 °C), producing necrosis in the patients’ surrounding tissues. To help avoid this problem, the addition of graphene could delay the polymerisation of the methyl methacrylate as it could, simultaneously, favour the optimisation of the composite material’s properties. In this work, we address the effect of different percentages of highly reduced graphene oxide with different wt.% (0.10, 0.50, and 1.00) and surface densities (150, 300, 500, and 750 m^2^/g) on the physical, mechanical, and thermal properties of commercial poly (methyl methacrylate)-based bone cement and its processing. It was noted that a lower sintering temperature was achieved with this addition, making it less harmful to use in surgery and reducing its adverse effects. In contrast, the variation of the density of the materials did not introduce significant changes, which indicates that the addition of highly reduced graphene oxide would not significantly increase bone porosity. Lastly, the mechanical properties (strength, elastic modulus, and fracture toughness) were reduced by almost 20%. Nevertheless, their typical values are high enough that these new materials could still fulfil their structural function. In conclusion, this paper presents a way to control the sintering temperature, without significant degradation of the mechanical performance, by adding highly reduced graphene oxide so that local necrosis of bone cement based on poly (methyl methacrylate) used in surgery is avoided.

## 1. Introduction

Bone cement, which consists of a broad range of different materials, is commonly used as a structural adhesive to fix replacement prostheses when the bone is damaged [1,2]. It has been used for biomaterial applications since the 1960s [3], and its central role is to fix implants by mechanical bonding and distribute loads optimally from the prosthesis to the bone. The implantation of prostheses has become an increasingly common practice throughout the world, both due to the rise and ageing of the population and the higher incidence of pathologies such as obesity [4], arthrosis (whose global prevalence is around 47% [5]), rheumatoid arthritis, osteoarthritis, and others. Therefore, the design of new optimised structural adhesives is an essential objective in traumatology and orthopaedics.

However, the limitation of biocompatibility and toxicity significantly reduces the variety of materials that can be used as adhesives or bone cement for prosthesis fixation. These materials are sintered directly in the operating room in two phases: (i) the solid phase, with poly-methyl methacrylate (PMMA) as the main bone cement compound, benzoyl peroxide (BPO) as the initiating agent, and barium sulphate (BaSO_4_) as a radio-opacifying agent to allow it to be detected on control radiographs; and (ii) the liquid phase, with the monomer methyl methacrylate (MMA), N, N-dimethyl-p-toluidine (DMPT) as the accelerator, and hydroquinone as the stabiliser or inhibitor of the polymerisation reaction.

After polymerisation of the bone cement, which usually takes a few minutes, residual MMA always remains in proportions of between 2–6%. However, this amount decreases with time, stabilising to around 1–2% of residual MMA after approximately one year [1]. This is because the mobility of the monomer exponentially decreases with the increasing viscosity of the bone cement. That MMA residue, together with small PMMA particles released from the bone cement during the service life, activates an inflammatory cascade where, after the rupture of the endothelium and vasoconstriction of the area, potentially necrotic ischemia of chemical origin occurs.

Osteoclasts, the only cells capable of resorbing bone [6], can respond to a wide variety of cytokines produced by cells of the innate and adaptive immune systems through highly specialised structures [7]. Subclasses of circulating monocytes, dendritic cells, and progenitor cells of the monocyte-macrophage line resident in the bone marrow can transform into osteoclasts if subjected to specific inflammatory signals [8,9,10]. Furthermore, the production and activation of T-lymphocytes cause a rise in osteoclast genesis and bone resorption since their interaction with the RANKL-RANK system increases osteoclast survival, delaying their apoptosis and making it possible for several cycles of osteoclastic activity [11]. All these inflammatory processes can lead to the loss of bone tissue, causing aseptic loosening, which accounts for approximately two-thirds of protheses’ revisions.

Graphene is a two-dimensional allotropic phase of carbon, ideally being a one-atom-thick layer, characterised by some exceptional properties: a large specific surface of 2.630 m^2^/g, an elasticity modulus of 1 TPa, a tensile strength of 130 GPa, great mechanical flexibility, excellent thermal conductivity, and nominal biocompatibility [12,13,14]. Therefore, adding graphene to commercial bone cement could improve the mechanical properties of the compound [15]. Additionally, the percentage of polymerisation from MMA to PMMA could be increased since graphene could inhibit the initiators, reducing the number of active polymerisation points, leading to slower but more efficient PMMA polymerisation. Significantly, highly reduced graphene oxide (HRGO) nanoplatelets are a derivative of graphene which are not entirely reduced and whose edges are highly reactive. Hydroxyl groups are formed on these limits, among other possible compounds, therefore acting as effective scavengers for polymerisation accelerators.

On the other hand, the MMA polymerisation reaction is highly exothermic, releasing 52–57 kJ/mol of MMA, resulting in a heat release of 1.4–1.7 × 10^8^ J/m^3^. Consequently, the bone cement is heated to high temperatures, between 70–120 °C. Thermal necrosis in tissues occurs at temperatures above 50 °C if exposed for more than one minute and 45 °C if exposure exceeds 30 min [16]. The minimum critical point of temperature to delay the death of osteocytes is around 47 °C, causing bone resorption, subsequent replacement, and medium and long-term disturbances in the anchorage of the implants [17,18,19,20,21,22]. Acrylic bone cement is in intimate contact with bone, and exposure to high temperatures can cause thermal necrosis in adjacent tissues. This can happen both due to direct cell damage from protein denaturation or through coagulation alterations that cause a lack of irrigation to the adjacent bone tissue, compromising the success of the surgery and accruing additional healthcare costs. Necrotic debris and tissues provide favourable conditions for bacterial growth and eventually lead to abscess formation.

The structure and vascularity of bone play an essential role in the response of bone tissue to heat. Cancellous bone dissipates heat faster and has a greater capacity for regeneration than compact (cortical) bone as it has a better supply of blood vessels. The addition of graphene to the bone cement could delay the polymerisation in a controlled way, and a reduction of the temperature rise and a longer workability time could be achieved by the surgeon.

Regarding mechanical properties, bone cement works well below its glass transition temperature, *T_g_*, which implies a brittle-elastic behaviour. Analysed from the point of view of geometric stability over time, the fact that bone cement works below *T_g_* is optimal, as the creep processes are negligible during the useful life of the bone cement, estimated at fifteen years. The lower value of the elasticity modulus of the bone cement facilitates a more homogeneous and even distribution of stress on the bone, thus avoiding stress concentrations that may damage it.

Furthermore, the addition of graphene to the adhesive could optimise the toughness or energy of fracture. The propagation of cracks in the matrix could be impeded or diverted by the presence of graphene because, locally, it would be more favourable to surround the nanoparticle than to fracture it. As the path of the crack increases and becomes more protracted and more tortuous, the energy necessary for the crack to grow and propagate increases [23,24,25].

Within this context, this work analyses the effect of the addition of different percentages of graphene (0.10%, 0.50%, and 1.00%) and several specific surface densities (150, 300, 500, and 750 m^2^/g) on the physical, mechanical, and thermal properties, as well as on the processing, of commercial PMMA-based bone cement.

## 2. Materials, Manufacture and Characterisation

In this study, a commercial bone cement widely used in traumatology was taken as the reference material. DePuy Ibérica S.L. (Madrid, Spain) supplied the starting excipients with CMW-1 Radiopaque reference, and their composition is shown in Table 1. The precursors are presented in two phases: (i) the solid phase, with poly-methyl methacrylate (PMMA) as the main bone cement compound, benzoyl peroxide (BPO) as the initiating agent, and barium sulphate (BaSO4) as the radio-opacifying agent; and (ii) the liquid phase, with the monomer methyl methacrylate (MMA), N, N-dimethyl-p-toluidine (DMPT) as the accelerator, and hydroquinone as the stabiliser or inhibitor of the polymerisation reaction.

To analyse the effect of graphene on the behaviour of the reference bone cement, four types of highly reduced graphene (HRGO) supplied by XG Sciences (Lansing, MI, USA), with different values for the specific area (m^2^/g) and proportions, were used. The characteristic values for these materials are shown in Table 2.

HRGO is characterised by its low oxygen content. However, the greater the surface area, the more difficult it is to reduce the oxygen content due to the high reactivity of the graphene nanoplatelet edge. For this reason, although in Table 2 the oxygen content increased with the specific area, they all correspond to the same HRGO manufacturing procedure.

To prepare PMMA compounds with HRGO, a high-resolution Mettler-Toledo 132 balance (up to 10^−6^ g) was used so that the percentage of each component could be determined with very high precision. To achieve a good dispersion of the graphene in the PMMA, once the solid phase of the precursor was mixed with the graphene, this combination was stirred in an ultrasound bath at 4.0 Hz for five minutes. After that, the liquid phase was added, and the exothermic reaction started. Subsequently, the entire mixture was spread on a 100 × 100 mm stainless steel mould with support bases coated with Teflon to avoid adhesion of the bone cement to the metal, making it easier to separate and more challenging to introduce undesirable mechanical deformations in the material. The amount of material deposited was calculated so that the resulting plate had a thickness of about 2.4 to 3.0 mm, the same thickness of the bone cement layer used in trauma operating rooms to fix the prosthesis to the bone.

Next, as shown in Figure 1, by employing an electromechanical testing machine (Instron 5866), a pressure of 50 kPa [25] was exerted perpendicular to the mould surface to homogenise the manufacturing conditions of the material and simulate the pressure that bone cement is subjected to through a finger or scalpel in the operating room [1,2,25,26,27,28].

Although the charge was initially maintained for 20 min to follow the preparation times indicated by the manufacturer, it was observed that the graphene content influenced the polymerisation time. Therefore, in some cases, it was necessary to maintain the charge for more than 24 h until the end of the polymer synthesis. Once the desired state of the bone cement was reached, the load was removed, the bone cement detached from the mould, and the material was stored in hermetically sealed PET zip bags at room temperature for one year before testing. Table 3 shows the specific composition and nomenclature used for each of the nine new manufactured materials.

### 2.1. Physical Characterisation

The density of each material was determined using the Archimedean immersion method in distilled water at a controlled temperature of 22.0 °C, using a Mettler-Toledo balance with an LC-P Density unit attached to measure the mass (Mettler-Toledo LLC, Columbus, OH, USA). The nominal dimensions of the test pieces were 26.8 × 3.0 × 2.4 mm^3^. Six tests were carried out for each composition, in which each one used five samples from different areas to avoid possible inhomogeneities in the developed materials.

### 2.2. Thermal Characterisation

Differential scanning calorimetry (DSC) was used to determine the glass transition temperature (*T_g_*) and the amount of residual monomer in the new materials. For this, a Mettler-Toledo DSC822e machine with STARe software was used. Samples of 6 mg were subjected to a heating ramp of 10 °C/min between 20 and 200 °C. To achieve cooling at the same speed, liquid nitrogen was used as a refrigerant.

### 2.3. Mechanical Characterisation: Tensile Test

To determine the tensile strength (*σ**_t_***) and the modulus of elasticity (***E***) of the new materials, simple tensile tests were carried out in an electromechanical testing machine with a ±1 kN load cell (resolution lower than ±0.1 N) and with an LVDT-type extensometer with a path of ±1 mm (resolution lower than ±1 µm). The tests were carried out in displacement control with a speed of 100 µm/min and an initial preload of 5.0 N. This preload was intended to achieve the perfect alignment of the specimen and ball joints of the loading device and thus avoid possible twisting or rotations of the sample, which would lead to inaccurate results.

The specimens for these tests were machined in the shape of a dog bone specimen with nominal dimensions of 10 mm testing length, 30 mm total length (*L_T_*), and 5 mm width at the heads inside the jaw; 25 mm centre shaft (*L*_0_), 3 mm at the thickness of the zone of the specimen with a reduced section (*A*), and 2.4 mm thickness (*B*). For each material composition, the samples, indicated in Table 3, were tested. To calculate the value of the maximum tensile strength (*σ_t_*), the following formula was used [29]:(1)$$\sigma =\frac{F_{m}}{AB}$$
where *F_m_* is the maximum force obtained during the tensile test, of which at least six tests were performed for each composition, and the mean value and its root mean square error were obtained.

### 2.4. Mechanical Characterisation: Fracture Toughness

The same mechanical testing instruments mentioned above were used, although three-point bending tests were performed on specimens notched with 0.13 mm diameter diamond wire to determine fracture toughness (*K_IC_*). In this case, the specimens had a preload of 1 N and were tested at a speed of 100 µm/min in the displacement control. This time, the experimental device consisted of a system of alumina rollers, 5 mm in diameter, with a span between the supports of 20 cm.

The nominal dimensions of the parallelepipedal specimens were: a length of 26.8 mm; width of 3.0 mm; and edge of 2.4 mm. As in the previous case, before each test, the dimensions of each specimen were measured with a calliper with a resolution of ±0.01 mm. The nominal length of the notches was 1.1 mm and was determined with a Nikon profile projector V-12B (Nikon, Leuven, Belgium), with a resolution of ±0.001 mm.

Finally, to calculate the fracture toughness from the maximum load and the dimensions of the specimens [30], the following equation was used:(2)KIC=σπa12Yah
where
(3)$$Y\left(\frac{a}{h}\right)=1.106-1.552\left(\frac{a}{h}\right)+7.71{\left(\frac{a}{h}\right)}^{2}-13.53{\left(\frac{a}{h}\right)}^{{}^{3}}+14.23{\left(\frac{a}{h}\right)}^{4}$$(3) is a geometric factor,
(4)$$\sigma =\frac{6M}{bh^{2}}$$(4) is the breaking stress,
(5)$$M=\frac{FL}{4}$$(5) is the maximum moment generated from the fracture force, *a* is the length of the notch, *b* is the width of the sample, *h* is the edge of the sample, *L* is the distance between supports, and *F* is the maximum fracture force.

### 2.5. Microstructural and Fractographical Analysis

For the analysis of microstructures and fracture surfaces of the studied materials, a Zeiss optical microscope and a high-resolution field emission scanning electron microscope (Auriga series, Carl Zeiss Microscopy LLC, White Plains, NY, USA) were used. To study the samples in the scanning electron microscope, they were fixed on a copper support with silver paint and were made conductive by depositing a nanometric layer (about 26 nm) of carbon in a LEICA EM ACE600 metalliser (Wetzlar, Germany). Again, at least six tests were performed for each composition and the mean value and its root mean square error were obtained.

## 3. Results and Discussion

Given the high number of experimental results obtained and the interrelationship between them, this section will show the results for each of the characterisations carried out, followed by a brief discussion. Finally, from the microstructural and fractographic analysis of the materials, an integrative discussion will present the macroscopical results obtained on the deformation and fracture micromechanisms observed at the microscale. This is intended to establish a correlation between microstructure and the properties that allow for the optimisation of future materials.

In all the physical and mechanical characterisations shown below, each experimental point corresponds to at least six measurements, and the error interval is the mean square error. In the thermal characterisation, only one measurement could be made for each experimental point according to a Gaussian statistic. This statistic is used for the analysis of the behaviour of materials in which there is no variability from one sample to another. As they are all assumed identical, the standard agreement in materials science and engineering expresses the mean values of the measurements made and their root mean square error. Evidently, this does not happen with living beings, and a different statistic must be applied since there is interindividual variation.

### 3.1. Physical Characterisation

Table 4 shows the variation of the density of each material, *ρ*, dependent on the percentage and surface area of the added HRGO. The displayed results show the mean value and root mean square error of at least six measurements. The error obtained in determining the density was, in almost all cases, less than 0.7%, which made it possible to detect minimal variations in density with the addition of HRGO.

Since the density of the graphene used was much lower than that of the PMMA, it was expected that the density of the composite material would be reduced. Therefore, when the graphene addition rates reached 1%, the expected reduction due to this effect using *the rule of mixtures* should be around this percentage. Nevertheless, the reduction in density due to the weight percentage of graphene was so slight that it was within the measurements’ error limits. In Table 4, increased wt.% led to a proportional reduction of the density.

Conversely, the possibility of HGRO platelets acting as pore generation points suggests that this could also reduce the final density. This effect can be appreciated in Table 4: the increase in the surface density of graphene tended to reduce the volumetric density of the compound, which would confirm the hypothesis that the greater the surface area of the HGRO, the more likely it is that nucleation of pores and inhomogeneities will occur around the platelets.

The only exception for these tendencies was G750-0.1, whose density was much lower than expected. This indicates that there was probably a problem during the synthesis of G750-0.1, which reduced the density by a more significant proportion than expected.

### 3.2. Thermal Characterisation

Table 5 shows the evolution of the glass transition temperature (*T_g_*) depending on the percentage and surface area of the added HRGO. As can be seen, this parameter was not visibly affected by the presence of HGRO, since random values appear around the value of PMMA without HGRO. Therefore, it can be considered almost independent of the presence of HRGO, at least in the proportions used and at long ageing times.

Something very different happened with the residual monomer, as can be seen in Table 6. A maximum of residual monomer was obtained with 300 m^2^/g and 0.5 wt.%, as indicated in Figure 2. This maximum was in the middle of the studied values, and there was a clear trend with a central maximum and a decrease in the residual monomer by increasing or decreasing the wt.% or specific area of the HRGO. This led us to believe that the two mechanisms may facilitate MMA polymerisation for long ageing times. HRGO is known to act as an inhibitor of the radical initiator of the polymerisation of MMA as it has a more significant amount of available double bonds compared to GO. Thus, an increase of the HRGO edge, by increasing the wt.% of the specific area, may result in a more significant number of double bonds and, therefore, a higher inhibition of the radicals initiating the reaction. Therefore, it is unlikely that the MMA encountered a radical polymerisation, obtaining a higher residual MMA, as shown by our results in the range of 0.1 to 0.5 wt.% and 120 to 300 m^2^/g. However, this does not explain the observations for high HRGO, 750 m^2^/g and 1.0 wt.%, where the residual MMA decreased in the long term. To explain this exciting result, we must consider the ease of MMA diffusion within the polymer itself. The simpler it is, the fewer polymeric chains there are in the polymer microstructure. If a more considerable amount of these radicals were inhibited by adding more significant amounts of HRGO, we would have reached a situation where there would have been fewer polymer chains, allowing them to slip easily and the MMA to reach one of the reactive points. Reducing the number of residual monomers is relevant from a clinical perspective, as the potential toxicity caused by MMA is well known [31].

In addition, the inhibition of radicals induced a slower polymerisation, thus reducing the rate of energy release, the escalation in temperature, and increasing the curing time, something which we observed in previous studies [28]. This effect would control the clinical needs of curing time and local temperature production by choosing the suitable percentage and density of HRGO. In this way, it would be possible to limit thermal damage to the surrounding tissues during surgery and increase the material’s working time according to surgical needs.

### 3.3. Mechanical Characterisation

From the simple tensile tests, the stress-strain curves of these materials were obtained, which, in all cases, showed macroscopically elastic and brittle behaviour until breaking. Therefore, the tensile strength, *σ_t_*, and the modulus of elasticity, E, were determined, as shown in Table 7 and Table 8, respectively. In addition, from the mechanical bending tests at three points on the notched specimens, the fracture toughness, *K_IC_*, was determined, which is shown in Table 9.

In all these experimental results, the increase of the HRGO contribution, either in percentage or in surface density, led to a degradation of all analysed mechanical properties, likely due to a decrease in the internal cohesion among the platelets and the PMMA matrix. This would justify that the resistance capacity of the compound was reduced significantly, even in tiny percentages, for the three analysed parameters. This phenomenon could become an inconvenience for the clinical application of these materials, as it is well known that fracture toughness is essential for fatigue life. However, the fact that the mechanical properties of the bone cement are inferior to those of the bone and the prosthesis guarantees that, in case of overload, the breakage is produced by this element, which is the most easily replaceable.

### 3.4. Microstructural and Fractographic Characterisation

The microstructural and fractographic analysis of the studied materials showed that, in some cases, large pores were trapped inside the material, as can be seen in Figure 3. Nevertheless, these kinds of pores are also typical in the reference bone cement.

In the same image, it can be appreciated that the fracture surface is relatively flat. The former may explain the abnormally low-density values of G750-0.1, while the latter is compatible with the brittle elastic behaviour observed macroscopically for these materials. Nevertheless, Figure 4 shows the typical porosity distribution, usual micrometric pores or smaller, and the barium sulphate radiopacifier appeared as a uniformly dispersed white contrast in the PMMA matrix.

During the detailed analysis of the fracture surfaces, poor adhesion of HRGO to bone cement (Figure 5 and Figure 6) and the existence of unwanted agglomerates of HRGO (Figure 7) were observed for all materials. Both factors led to the creation of cracks and defects around the HRGO platelets.

A more significant number of cracks increases the likelihood of them reaching the critical crack size and propagating catastrophically under less stress than the expected critical value. For example, Figure 7 shows an HRGO lamella utterly parallel to the fracture surface. The gap between the bone cement and that HRGO lamella, or between HRGO lamellae, could be the place where crack propagation started in this case. Consequently, both factors indicated that the presence of the HRGO in the PMMA matrix justified the macroscopic degradation of the mechanical behaviour of the composite observed in the previous section.

## 4. Conclusions

From the experimental observations shown above and their discussion, the following conclusions can be established:After one year of natural ageing, the density and glass transition temperature were not affected by the addition of different percentages and surface densities of HGRO;The amount of residual MMA varied with the amount of HRGO and its specific area. This phenomenon was related to reducing the polymerisation rate in the presence of HRGO and MMA diffusivity inside the PMMA. It would allow for controlling the curing time and the local temperature increase, which would help limit the thermal damage caused to the surrounding tissues during surgery and increase the working time of the material if necessary;Due to the addition of different percentages and surface densities of HGRO, the analysed mechanical properties suffered an intense degradation, exceeding 40%;To improve the mechanical behaviour of these compounds, it is essential to find micromechanisms that increase the cohesion between the HRGO and the PMMA matrix, like the silanisation of graphene;Furthermore, we would like to point out the limitations of this study. It was not a clinical investigation aimed at immediate application in patients, but a scientific analysis under laboratory conditions to determine the new composite materials’ potential;As future work, it would be fascinating to carry out this type of study on other bone cement to which similar proportions of HRGO are added and compare the composites’ results with similar proportions of reduced graphene and different bone cement matrices. Looking ahead, it would be interesting to extend this study to lower concentrations of HGRO, consider the influence of HGRO on the bacterial colonisation of cement, analyse the evolution of the physical, thermal, and mechanical behaviour of bone cement with graphene after its controlled immersion in physiological serum, and study the influence on antibiotic release when added to bone cement.

## Figures and Tables

**Figure 1 polymers-13-02027-f001:**
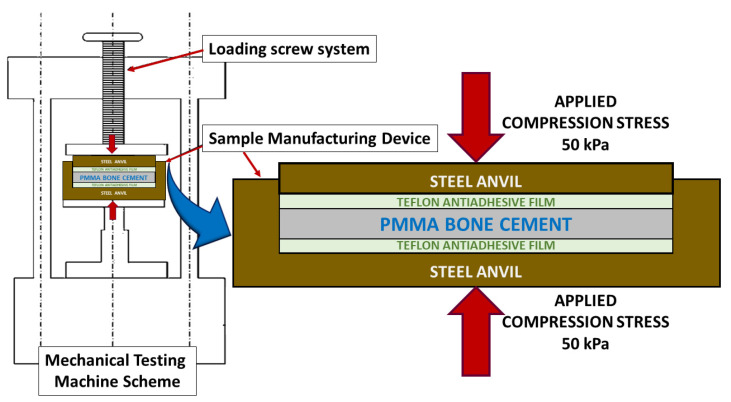
Simplified transversal section of the device developed for controlled fabrication of the bone cement. On the right, enlarged, it can be seen in greater detail transversal section detail of the sample manufacturing device.

**Figure 2 polymers-13-02027-f002:**
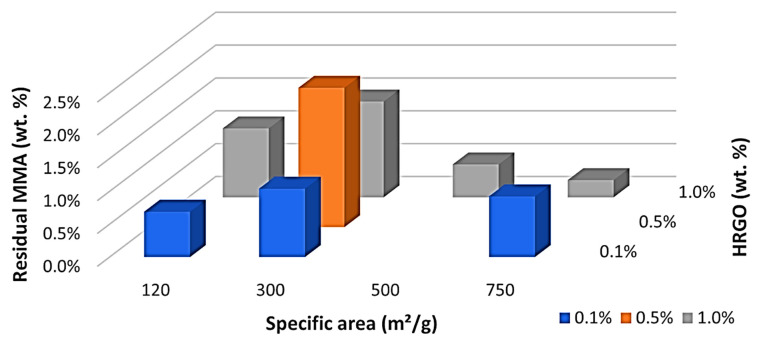
Residual monomer (MMA) versus specific area and percentage (blue = 0.1%, orange = 0.5%, and grey = 1.0%) by weight of HRGO.

**Figure 3 polymers-13-02027-f003:**
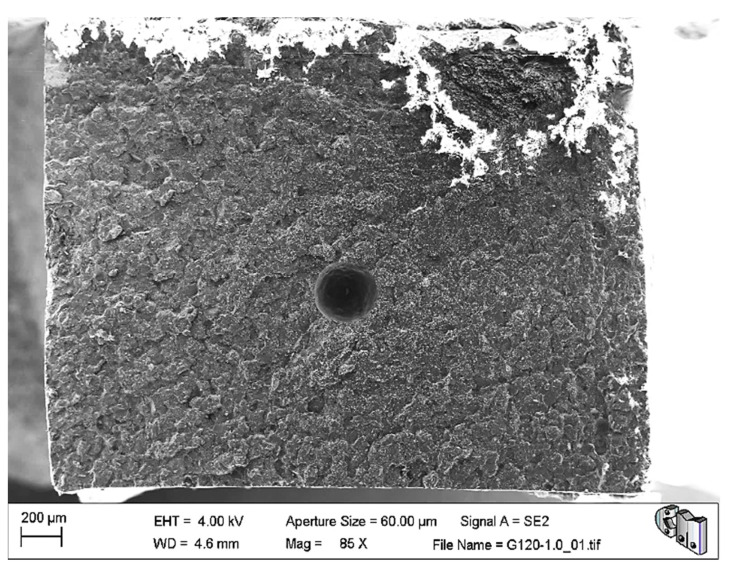
SEM image of the tensile fracture surface of G120-0.1. A giant air bubble can be seen trapped within the bone cement.

**Figure 4 polymers-13-02027-f004:**
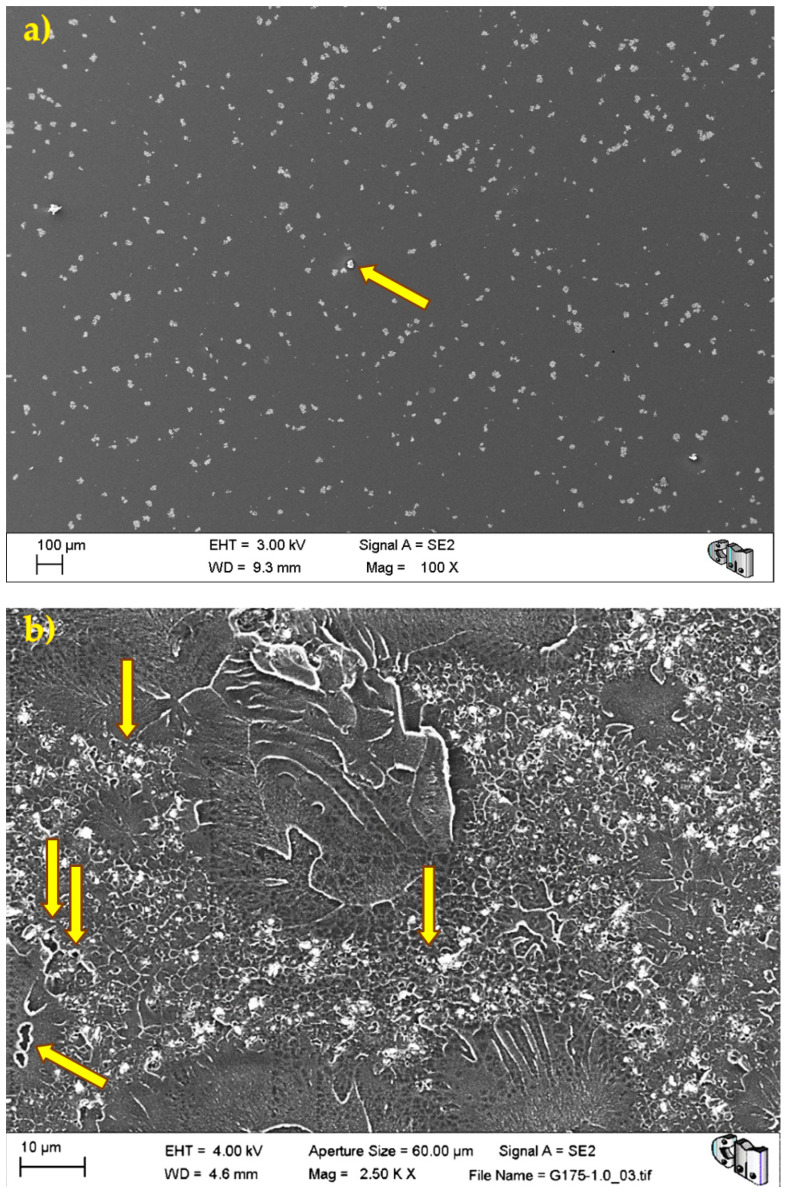
Microstructure of G120-0.1: (**a**) at low magnification, the barium sulphate radiopacifier appears as a uniformly dispersed white contrast in the PMMA matrix; and (**b**) at higher magnification, some tiny pores and defects can be observed.

**Figure 5 polymers-13-02027-f005:**
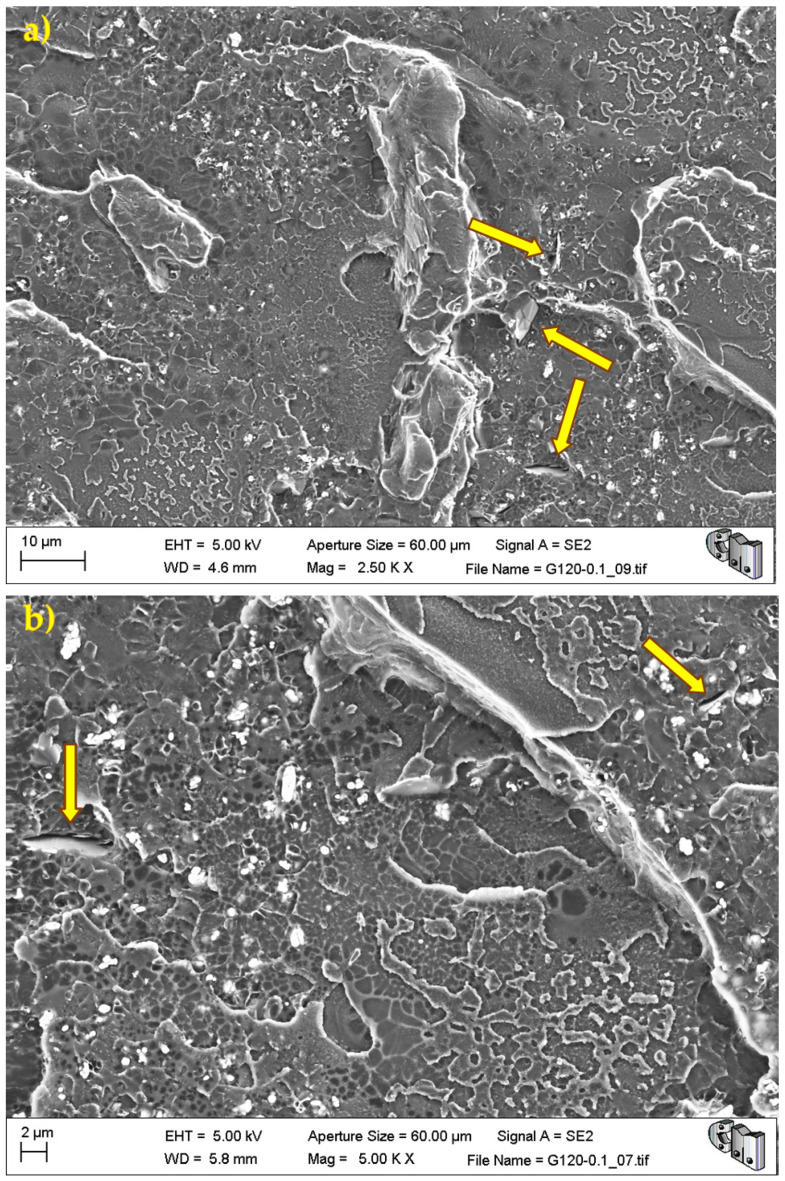
G120-0.1 fracture surface. The arrows indicate voids and flaws between the bone cement and the HRGO due to poor adhesion between both: (**a**) lower magnification, and (**b**) higher magnification.

**Figure 6 polymers-13-02027-f006:**
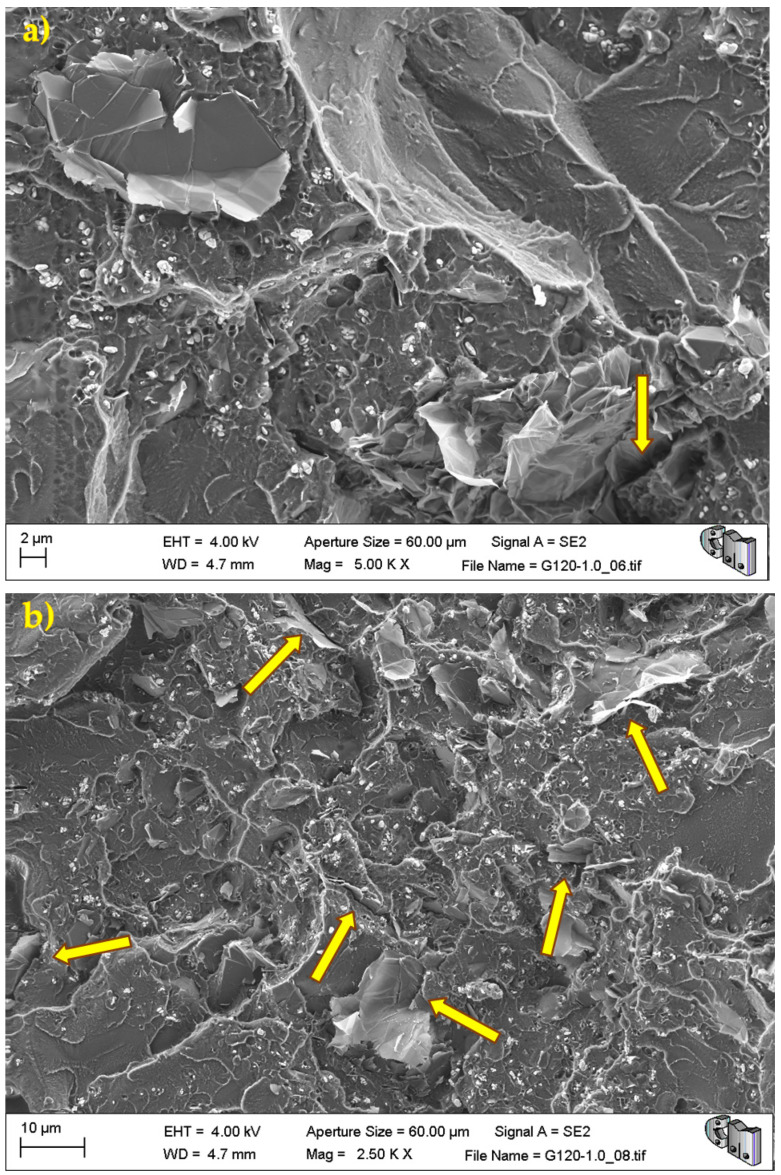
G120-1.0 fracture surface. The arrows indicate voids and flaws between the bone cement and the HRGO due to poor adhesion between both: (**a**) higher magnification, and (**b**) lower magnification.

**Figure 7 polymers-13-02027-f007:**
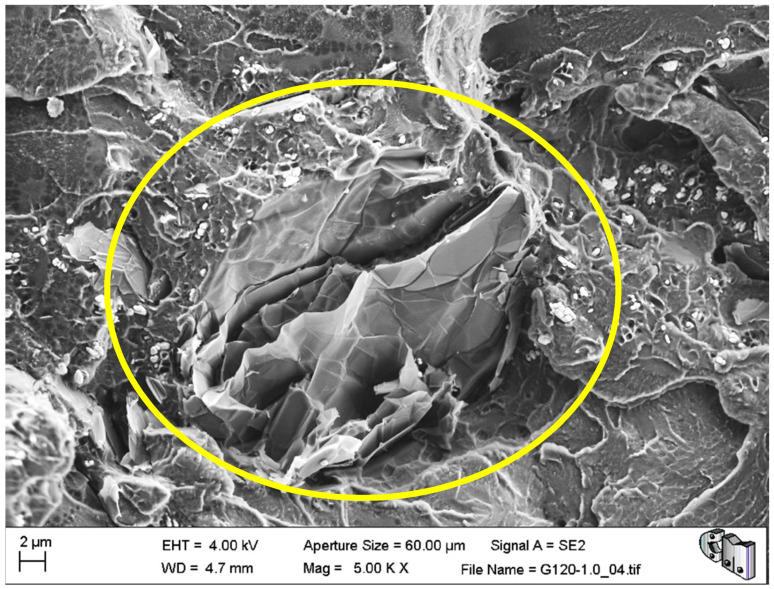
G120-1.0 fracture surface with a HRGO agglomerate perpendicular to the fracture surface.

**Table 1 polymers-13-02027-t001:** Composition of DePuy Ibérica CMW 1 bone cement radio-opaque.

Solid Phase			Liquid Phase		
Compound	Proportion	Function	Compound	Proportion	Function
**Poly-methyl methacrylate (PMMA)**	88.85 wt.%	Polymer	**Methyl methacrylate (MMA)**	99.18 wt.%	Monomer
**Benzoyl peroxide (BPO)**	2.05 wt.%	Initiating agent	***N*** **,*N*-dimethyl-p-toluidine (DMPT)**	≤0.82 wt.%	Accelerator
**Barium sulphate (BaSO_4_)**	9.1 wt.%	Radio-opacifying agent	**Hydroquinone**	25 ppm	Stabilizer

**Table 2 polymers-13-02027-t002:** Composition of XG Sciences highly reduced graphene oxide (HRGO).

Graphene	Specific Area (m^2^/g)	Atomic Oxygen Concentration (%)	Mean Diameter (µm)	Density (g/cm^3^)
**HRGO 120**	120–150	<1	15	0.03–0.10
**HRGO 300**	300	5	<2	0.20–0.40
**HRGO 500**	500	7	<2	0.20–0.40
**HRGO 750**	750	10	<2	0.20–0.40

**Table 3 polymers-13-02027-t003:** Nomenclature of bone cement composition with HRGO manufactured in the form of 100 × 100 × 5 mm^3^ plates.

	Surface Area of the Graphene Used (m^2^/g)			
HRGO Weight Percentage (wt.%)	120	300	500	750
0.1	G120-0.1	G300-0.1	-	G750-0.1
0.5	-	G300-0.5	-	-
1.0	G120-1.0	G300-1.0	G500-1.0	G750-1.0

**Table 4 polymers-13-02027-t004:** Material density versus the percentage and surface area of the added HRGO. Each result shows the mean value and root mean square error.

	*ρ* (g/cm^3^)			
	Surface Area of the Graphene Used (m^2^/g)			
HRGO Weight Percentage (wt.%)	120	300	500	750
0.0	1.244 ± 0.002 [25]			
0.1	1.240 ± 0.004	1.240 ± 0.002	-	1.182 ± 0.007
0.5	-	1.229 ± 0.009	-	-
1.0	1.241 ± 0.003	1.236 ± 0.003	1.230 ± 0.006	1.234 ± 0.004

**Table 5 polymers-13-02027-t005:** Glass transition temperature (*T_g_*) versus the percentage and surface area of the added HRGO.

	*T_g_* (°C)			
	Surface Area of Graphene Used (m^2^/g)			
HRGO Weight Percentage (wt.%)	120	300	500	750
0.0	69.6 [17]			
0.1	70.2	69.8	-	70.2
0.5	-	68.0	-	-
1.0	68.3	69.3	70.8	70.3

**Table 6 polymers-13-02027-t006:** Residual monomer MMA (%) versus the percentage and surface area of the added HRGO.

	Residual MMA (%)			
	Surface Area of Graphene Used (m^2^/g)			
HRGO Weight Percentage (wt.%)	120	300	500	750
0.1	0.67	1.04	-	0.92
0.5	-	2.05	-	-
1.0	1.05	1.67	0.50	0.26

**Table 7 polymers-13-02027-t007:** Tensile strength, *σ_t_*, of the material versus the percentage and surface area of the added HRGO. Each result shows the mean value and root mean square error.

	*σ*_t_ (MPa)			
	Surface Area of the Graphene Used (m^2^/g)			
HRGO Weight Percentage (wt.%)	120	300	500	750
0.0	38 ± 2 [25]			
0.1	39 ± 7	36 ± 4	-	37 ± 4
0.5	-	31 ± 5	-	-
1.0	30 ± 5	29 ± 3	29 ± 5	20 ± 1

**Table 8 polymers-13-02027-t008:** Modulus of elasticity, *E*, of the material versus the percentage and surface area of the added HRGO. Each result shows the mean value and root mean square error.

	*E* (GPa)			
	Surface Area of the Graphene Used (m^2^/g)			
HRGO Weight Percentage (wt.%)	120	300	500	750
0.0	3.20 ± 0.02 [25]			
0.1	3.2 ± 0.6	3.2 ± 0.2	-	3.0 ± 0.2
0.5	-	2.8 ± 0.3	-	-
1.0	3.1 ± 0.1	2.8 ± 0.3	2.3 ± 0.2	2.2 ± 0.3

**Table 9 polymers-13-02027-t009:** Fracture toughness, *K_IC_*, of the material versus the percentage and surface area of the added HRGO. Each result shows the mean value and root mean square error.

	*K_IC_* (MPa·m^1/2^)			
	Surface Area of the Graphene Used (m^2^/g)			
HRGO Weight Percentage (wt.%)	120	300	500	750
0.0	1.28 ± 0.02 [25]			
0.1	1.40 ± 0.02	1.20 ± 0.05	-	1.15 ± 0.05
0.5	-	1.12 ± 0.02	-	-
1.0	1.08 ± 0.03	0.99 ± 0.04	1.06 ± 0.05	0.94 ± 0.06

## Data Availability

Data is contained within the article.

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
