# Peer review of "Influence of HRGO Nanoplatelets on Behaviour and Processing of PMMA Bone Cement for Surgery"

_polymers, 2021, doi:10.3390/polym13122027_

Round 1
Reviewer 1 Report
The authors have presented a good work. However, the manuscript is full of conceptual and methodological errors that the authors should resolve for publication:
- First of all, the title is attractive. However, it does not correspond to the reality and the conclusions of the manuscript presented by the authors. Authors should briefly review the title, and propose a change.
- The abstract presented by the authors is too cumbersome. More details should be provided, defining abbreviations at all times. Authors should add a conclusion in this section.
- The keywords submitted by the authors are too many. The reader is lost with abbreviations. This point should be reviewed. Authors must include general keywords and some specific ones. Authors must understand that the reader is looking for these keywords when he consults the information.
- A graphic summary is necessary in this manuscript.
- The introduction needs to be improved substantially. For example: appointment 1 must be changed to some other specific one. Cite articles or reviews. Also, the statement on lines 28-29 is not true.
- In the introduction the authors must realize and focus the state of the art. The authors are making a meaningless digression, where they do not focus on current cement formulations, as well as the problems that current ones have (infections, etc .....). All this makes it possible to justify the spirit of the study. Right now, these points are not being answered, nor are they being taken into consideration.
- In the introduction the authors must realize and focus the state of the art. The authors are making a meaningless digression, where they do not focus on current cement formulations, as well as the problems that current ones have (infections, etc .....). All this makes it possible to justify the spirit of the study. Right now, these points are not being answered, nor are they being taken into consideration.
8.Line 94-100 is necessary appointments.
- The Materials, Manufacture and Characterization section must be detailed. Authors must include references and detailed information.
- In the first place it is necessary that the authors make an exhaustive description of the composition of the material. This point should be included in the supplementary material.
- Table 2 gives the Mean Diameter, the authors must give the interquartile ranges so that the reader can have an idea of the deviations.
- How do the authors justify that HRGO 750 has an Atomic oxygen concentration of 10%? and that the specific area can vary so much in HRGO 120?
- Line 132 (n Mettler-Toledo 132
balance) describe exhaustively.
- Check from line 132 to 143. Missing information. Authors should create an explanatory figure.
- Figure 1 needs to be revised, it is not understood and is too poor. The authors must define in the figure legends all the terms used. The authors must justify the pressure.
- Line 158, the authors say they specify the composition. The reviewer does not see it, what is presented by the authors is too limited.
- Point 2.1 please expand the information and provide the data in the text or in supplementary material.
- Point 2.2. provide the data. Why 6 mg? Justify.
- Describe temporally and causally the tensile tests to arrive at your equations.
- The fracture is not calculated this way, please review the model and provide specific references.
- The Methodology of Scanning Electron 217
microscope is not described.
- Statistics are missing. Authors must statistically compare their data, this is essential to support their conclusions.
- The results are not presented in a disputed way as the authors intend. Paragraphs are presented where no references are provided, making an intention more than discussion. The authors must describe their results and relate it to what exists.
- Figure legends should contain more information. Abbreviations must be described. Authors must provide data.
- Table 4 should have the mean with the interquartile data and a statistic. All this is applicable to the rest of the tables.
- Figure 2 is of very poor quality. Provide a well-made graph with a statistical program and its error bars, describing and discussing the information.
- Figures 3 to 7 should be better described. The indicated information should be increased and give information about the arrows that the authors point to. Please describe this information in more detail in the text.
- The authors do not provide actual control groups, or any other type of already commercial material. For this reason, I suggest that the authors should provide a final table where they compare the properties that their new formulations provide with the existing ones. The authors must describe other cements and thus endorse their conclusions and the significance of their cement.
- The authors must provide their innumerable limitations.
- The authors should describe some more elaborate conclusions, where the authors refer to the translational character and the importance that the implementation of their new formulations may have in clinical practice and in translational and precision medicine.
- Authors should provide a representative outline of the material for a better understanding of their findings.
- There are many grammatical errors. Authors should review their grammar with an expert.
Author Response
Dear reviewer, thank you very much for your detailed insights and in-depth analysis of our work, which have contributed significantly to its improvement. Attached we answer each of your questions.

Reviewer 2 Report
Dear Authors
If possible try to change the title of the manuscript or rewrite it. It's very difficult and will not visible in many search strategies applied by the researchers.
- An abstract is fine just check the flow of the work.
- The introduction needs editing.
- First line of the introduction is not correct. Bone cement is not always polymeric form. Rewrite first line of the introduction. Also, below reference is latest paper on the topic authors selected and help them to improve introduction part of this paper.
a) Zhao, R.; Yang, R.; Cooper, P.R.; Khurshid, Z.; Shavandi, A.; Ratnayake, J. Bone Grafts and Substitutes in Dentistry: A Review of Current Trends and Developments. Molecules 2021, 26, 3007. https://doi.org/10.3390/molecules26103007
Authors did not reference properly. " adaptive immune systems [6] [7] [8]". Merge them in one bracket in the whole manuscript.
Line 88-93: Adjust these line in the introductory paragraph not suitable in this place.
This section needs serious attention from authors: Materials, Manufacture and Characterisation. They write in a rough style. Need scientific writing here.
Is any trademark of these materials DePuy Ibérica CMW 1 bone cement?
Figure-1: need high resolution.
The result and discussion heading have many coherence issues. Statements continuation and flow is not maintained by the authors. It's better to rewrite it properly.
Authors have to write few lines about the limitations of this work.
Author Response

(The authors gave the same response as above.)

Round 2
Reviewer 1 Report
After reading with interest the new version of the authors, the reviewer should report very unfavorably. Authors should read the reviewer's comments and not avoid them.
The authors have only made minor changes to their article. the changes are not marked and the answers are not adequate. A very clear example is figure 1, only the authors have put it in color.
Images are now inconsistent and unevenly sized. The authors have not made an adequate modifying of the discussion.
The article is deficient in its current form.
Author Response
Dear Reviewer:
Thank you very much for your suggestions. Attached I send you our final version of the paper that has been evidently improved thanks to your revisions.
My best personal regards and health wishes for all,
Jose Ygnacio Pastor
Prof. Dr. in Materials Science and Engineering
Technical University of Madrid

Reviewer 2 Report
Good improvements
Author Response

(The authors gave the same response as above.)
